# Comparison of the Time to Extubation and Length of Stay in the PACU after Sugammadex and Neostigmine Use in Two Types of Surgery: A Monocentric Retrospective Analysis

**DOI:** 10.3390/jcm10040815

**Published:** 2021-02-17

**Authors:** Cyrus Motamed, Jean Louis Bourgain

**Affiliations:** Department of Anesthesia, Institut Gustave Roussy, 94080 Villejuif, France; bourgainjl@orange.fr

**Keywords:** neostigmine, sugammadex, residual paralysis, length of stay in recovery

## Abstract

Sugammadex provides a rapid pharmacological reversal of aminosteroid, as well as fewer pulmonary complications, a better physiological recovery, and shorter stays in the postanesthetic recovery unit (PACU). This retrospective analysis of our Centricity anesthesia database in 2017–2019 assessed the efficiency of sugammadex in real-life situations in two groups of surgical cancer patients (breast and abdominal surgery) based on the extubation time, operating room exit time, and length of PACU stay. Overall, 382 anesthesia records (131 breast and 251 abdominal surgeries) were extracted for the pharmacological reversal of neuromuscular blockades by neostigmine or sugammadex. Sugammadex was used for reversal in 91 breast and 204 abdominal surgeries. Sugammadex use did not affect the extubation time, operating room exit time, or length of PACU stay. However, the time to reach a 90% train of four (TOF) recovery was significantly faster in sugammadex patients: 2 min (1.5–8) in breast surgery and 2 min (1.5–7) in abdominal surgery versus 10 (6–20) and 9 min (5–20), respectively, for neostigmine (*p* < 0.05). Most patients who were reversed with sugammadex (91%) reached a TOF ratio of at least 90%, while 54% of neostigmine patients had a 90% TOF ratio recorded (*p* < 0.05). Factors other than pharmacological reversal probably influence the extubation time, operating room exit time, or PACU stay; however, sugammadex reliably shortened the time so as to reach a 90% TOF ratio with a better level of reversal.

## 1. Introduction

Sugammadex is the latest neuromuscular reversal agent developed for aminosteroids, mostly rocuronium, and has a significantly faster recovery in comparison to neostigmine [1]. Sugammadex encapsulates the aminosteroid agent to create water-soluble complexes [2]. By contrast, neostigmine acts as a competitive inhibitor of acetylcholinesterase and prolongs the presence of acetylcholine in the neuromuscular junction, which ultimately antagonizes the neuromuscular blocking agent and restores neuromuscular function [2]. Sugammadex is also characterized by its capacity to reverse the paralysis of a deep neuromuscular blockade within a short time [3,4,5]. Multiple studies have described a significant positive outcome in terms of operating room discharge, time to extubation, postoperative pulmonary recovery, quality of postoperative recovery, and speed of discharge from operating room to the postanesthetic recovery unit (PACU); however, these data are sparse and sometimes inconclusive [6,7,8].

In our department, monitoring the quantitative neuromuscular blockade is mandatory as part of our quality assurance program in anesthesia [9,10]. Specific and simple local guidelines, based on the level of paralysis, suggest pharmacological reversal of rocuronium, performed with either neostigmine or sugammadex, with their dosage given in a rational manner [11]. In this retrospective analysis, using our Anesthesia Information Management System (AIMS), we assessed two types of surgery (breast and general surgery for cancer) in which the pharmacological reversal of rocuronium muscle paralysis was achieved by neostigmine or sugammadex. We hypothesized that the fast recovery of neuromuscular blockade after sugammadex might affect the extubation time and length of stay of patients in the PACU differently, depending on the type of surgery with different durations. Our primary objective was to compare the time to extubation, time to exit the operating room, and length of stay in the PACU. In addition, we also assessed the neuromuscular parameters such as monitoring the train of four (TOF) ratio at the time of the reversal injection and during the process of extubation.

## 2. Methods

This study was a retrospective database analysis, and no informed consent was necessary since no change in practice was required. We have authorization from our institutional review board to retrospectively exploit our GE Centricity anesthesia database (Barrington, IL, USA).

The database was queried by a sequential query language method (SQL) through Crystal report software V8. The anesthesia database administrator (J.L.B.) underwent several cycles of formal training in order to master the appropriate data extraction from the database. Each broad extraction was verified by a preliminary extraction of the same query through a very short period (three days) and checked manually to validate the query before extracting the targeted data for the broader time period. Data were cleaned for redundancy, and missing data were calculated using Excel 2010 spreadsheets. The outliers were checked on the anesthesia reports and were corrected or omitted. The missing data (weight and height, for example) were extracted from the patient’s medical report.

The studied population consisted of breast and abdominal surgery patients from 1 January 2017 through 31 July 2019 in order to provide a homogenous group of patients. The local guidelines for the pharmacological reversal of rocuronium are as follows: neostigmine (20 or 40 µg/kg) if four responses are available, depending on the TOF ratio, and sugammadex (2 or 4 mg/kg) depending on the first response to TOF stimulation or post tetanic count stimulation.

The following parameters were collected: demographic characteristics, type of surgery, body temperature at arrival in the PACU, rocuronium dosage, the time of the first and last modality of injection of rocuronium, the type and dosage of pharmacological reversal (neostigmine or sugammadex), and the type and quantity of inhalational or intravenous anesthetic agent. As part of our quality assurance program, all operating rooms are equipped with the same neuromuscular monitoring device integrated into the main vital sign monitoring system (Philips MX 800^®^ with an accelerometer), which is itself connected to our Centricity anesthesia database. The intensity of the ulnar nerve stimulation was automatically calibrated before the muscle relaxant injection. Since this was a retrospective assessment, anesthesia protocols were not standardized and were at the discretion of the anesthetist in charge. However, they always consisted of propofol IV bolus and remifentanil target-controlled infusion (RTCI) for the induction and were adjusted for maintenance in addition to inhalational anesthesia or propofol IV, also taking into consideration the presence or not of regional anesthesia (epidural for abdominal surgery and paravertebral block for breast surgery). At skin closure, morphine IV 0.1 to 0.15 mg/kg was injected, again in accordance with the presence or not of a regional anesthetic technique or infiltration. The postoperative nausea and vomiting protocol is standardized in our institution by intraoperatively using dexamethasone and ondansetron, followed by droperidol IV if necessary in the PACU. There are no specific criteria for the transfer of a patient from the operating room to the PACU; however, all patients need to be extubated under a stable neurologic, hemodynamic and respiratory status unless there is a direct transfer to the intensive care unit with respiratory or hemodynamic support. Patients are discharged from the PACU after checking for an acceptable modified Aldrete score including a verbal pain and sedation score lower than 2, central temperature >36.5°C and stable hemodynamic and respiratory conditions and no postoperative nausea and vomiting. Therefore, the following parameters were extracted: the presence of neuromuscular monitoring and data such as calibration and the TOF ratio before and after reversal and before extubation. These measurements are always performed in the operating room at the time of extubation (except in patients who are directed to the ICU for postoperative ventilation). The time of the last End Tidal (ETCO_2_), the time of extubation, the central temperature upon arrival, the five-scale sedation score, and the length of stay in the PACU were recorded.

### Statistical Analysis

The sample sizes were chosen with regard to the results of previous investigations [12] assuming an average of a 20% difference in the incidence of residual paralysis between neostigmine and sugammadex by using a sample size calculator for a comparison of proportions in MedCalc^®^ version 14 (Ostend, Belgium); the minimum study participants would be 40 for neostigmine and 80 for sugammadex with a power of 95% and α error set at 5%. The normally distributed data are presented as the mean and standard deviation (or minimum and maximum when appropriate), and the categorical data are presented as percentages when appropriate. A Student’s *t* test and chi square test were used to compare data. MedCalc^®^ software was also used to analyze the data.

## 3. Results

The demographic and surgical characteristics are presented in Table 1 and Table 2. No significant differences were noticed between patients except for emergency abdominal surgery patients. All patients arrived normothermic in the PACU. The mean duration of surgery was significantly longer in the abdominal surgery group (270 ± 120 min) than in the breast surgery group (160 ± 120 min) (*p* < 0.05). The length of stay in the PACU was not significantly different between reversals or between the surgery group patients, with a mean time of 97 ± 48 min.

Sevoflurane was used in 68% of the patients, and Desflurane was used in the others (*p* < 0.05). The use of propofol for maintenance was less than 2%. The calibration procedure for the NMT Philips MX800 accelerograph was noted without error in 65% of the breast surgery patients and in 72.5% of the abdominal surgery patients. Overall, eight patients (two in breast surgery and six in abdominal surgery) did not have any record of neuromuscular monitoring. The mean dose of rocuronium (bolus only) was 53 ± 36 mg in the breast surgery group and 85 ± 54 mg (bolus + infusion) in the abdominal surgery group. In these patients, rocuronium was either injected by bolus or by infusion in the case of robotic surgery.

The mean dose of sugammadex was 3.3 ± 2.14 mg/kg in the breast surgery group and 4.5 ± 3.5 mg/kg in the abdominal surgery group. The mean dose of neostigmine was 34 µg/kg for all patients. The mean time to achieve 90% recovery was 9 min (5–20) for neostigmine and 2 min (1.5–7) for abdominal surgery.

At the time of extubation, in the abdominal group, when sugammadex was used, all patients had T4/T1 > 100%. In the breast surgery group, only 67% had a T4/T1 > 90% (*p* < 0.05) for the reversal. For neostigmine, the percentage of T4/T1 < 90% was 35% for breast surgery patients and 42% for abdominal surgery patients (*p* > 0.05) (Table 3).

The time to extubation after the injection of the pharmacological reversal was similar in all groups (Table 3). The time to the operating room exit and transfer to the PACU was similar in each type of surgery, being an average of 4 min longer for abdominal surgery patients (Table 3).

## 4. Discussion

This study showed, as expected, that using sugammadex instead of neostigmine did reduce the waiting time so as to reach a 90% TOF recovery. However, in our series of patients, it did not decrease the time to extubation, time to exit the operating room, or length of stay in the PACU.

Carron et al. [7] determined that sugammadex reduced the readiness time and time to discharge both from the operating room and the PACU for morbidly obese patients. In the present study, we could not demonstrate an accelerated exit from the operating room to the PACU. However, this study was a real-life retrospective analysis, and we did not assess the specific BMI, as we do not have a high percentage of patients who are morbidly obese. In addition, the real time to extubation calculated for each type of reversal might not be totally reliable because, in some patients, the anesthesia providers could have deliberately waited longer in order to avoid the use of sugammadex. Moreover, it is possible that the time to extubation was more influenced by the awakening rather than the muscle relaxant’s recovery, as the expired concentrations of inhalational anesthetics were relatively high.

This study highlights several secondary findings. One is that despite the presence of guidelines for rational pharmacological reversal, some aberrant uses of sugammadex (especially the dosage) are noted. For example, we suspect that once a sugammadex vial is opened, the anesthesia providers might consider finishing it in order not to “waste an expensive product.” Moreover, we also speculate that the durations of surgeries were also taken into consideration by the attending anesthesiologist when choosing between sugammadex and neostigmine. In addition, our study highlights the importance of neuromuscular monitoring; for example, even in cases of intended sugammadex reversal, sugammadex might sometimes not be necessary and recovery could be obtained spontaneously or with neostigmine [11,13]. Furthermore, on occasion a small percentage of patients did not have adequate monitoring recording because of an initial error in calibration or a failure to use the monitoring device. In addition, a very small number of patients (2%) did not have any trace of monitoring without any plausible explanation in the record; however, a putative technical problem cannot be ruled out.

A better physiologic recovery after sugammadex was reported by Kim et al. [14]; however, we did not search for this positive outcome as it is not an item that is collectable from our database. Nevertheless, if it existed, it did not affect our time to discharge from the PACU. Indeed, a better physiologic outcome might not always reflect a shorter discharge time from the PACU. In addition, Kim et al. [14] assessed the effect of a single bolus of rocuronium in a randomized controlled trial, while our study, as a database analysis in real life, considered all types of rocuronium injections, bolus reinjections, and infusion administrations [14].

A recent observational matched cohort study reported that sugammadex administration was associated with a 30–55% reduced risk of major pulmonary complications when compared with neostigmine [15,16]. We did not search for this complication, mainly because of a lack of power which did not permit us to extrapolate our results to other studies. As reported, we did not detect an acceleration of the time to extubation, despite a significant improvement in achieving a 90% recovery in patients receiving sugammadex. The time to extubation is different from the time to TOF recovery, as extubation depends on several other parameters, including respiratory parameters, neurologic cooperation, hemodynamic stability, and the central temperature.

The time to exit the operating room was not significantly affected by the use of sugammadex, as many factors other than the reversal of the neuromuscular blockade are involved, such as the presence of orderlies and the PACU’s capacity to take charge of a new patient. Nevertheless, 91% of the patients who were monitored and reversed with sugammadex had a TOF ratio over 90%, which is an important safety and reliability issue, as previously reported [17]. Our results somewhat confirm previous findings that the use of intraoperative quantitative NMB monitoring and sugammadex is probably associated with a lower incidence of residual paralysis [18] in comparison to NMB monitoring and neostigmine. To avoid residual paralysis, it is necessary to monitor all patients until the extubation time regardless of the administration of reversals. This is true before administration and before extubation, using different criteria depending on which reversal is used.

The shortcomings of this study were its retrospective nature, the small sample size, and the limit of the types of surgery to only two. We believe that generalizing our findings to the general population will be difficult before further prospective studies in these fields are conducted. However, regular quality assurance interventions should improve these results.

## 5. Conclusions

This study suggests that, despite a total efficiency and a significant reduction in time to reach a 90% TOF ratio with sugammadex, the time to extubation, time to exit the operating room, and the length of stay in the PACU were not significantly reduced in either breast or abdominal surgeries when compared with neostigmine. However, since this was a single-center, retrospective, nonrandomized analysis, the results cannot be generalized but could be used to at least improve some of our practice.

## Figures and Tables

**Table 1 jcm-10-00815-t001:** Demographic characteristics for breast surgery patients.

	Breast Surgery (*n* = 131)		Significance *p* Value
	Neostigmine (*n* = 40)	Sugammadex (*n* = 91)	
ASA (1/2/3/4/E) (%)	6/26/1/0/3 (15/65/2/0/8)	15/67/9/0/3 (16/73/8/0/3)	0.1
Age (mean ± SD)	54 ± 14	58 ± 14	0.13
Height (mean ± SD)	163 ± 6	163 ± 7	1
Weight (mean ± SD)	72 ± 18	76 ± 20	0.2
Hypnotics DES/SEV/PROP (%)	10/28/2 (25/70/5)	24/64/3 (25/69/3)	0.8
ET inhalational agent (at the time of extubation) (mean ± SD)	DES 0.92 ± 0.35 SEV 0.24 ± 0.15	DES 0.73 ± 0.36 SEV 0.22 ± 0.17	0.05 0.2
Duration of surgery (minutes) (mean ± SD)	184 ± 120	146 ± 108	0.07
Temperature (°C) upon arrival in PACU (mean ± SD)	36.5° ± 0.7°	36.4° ± 0.5°	0.35
Sedation score upon arrival in PACU (0/1/2/3/4) (%)	68/26/5/1/0	71/23/5/1/0	0.9
Length of stay in PACU (minutes) (mean ± SD)	106 ± 44	106 ± 60	1

DES = desflurane, SEV = sevoflurane, PROP= propofol, E= Emergency, ET = End tidal, Sedation scores on a 5-grade scale (0 = alert; 1 = occasionally drowsy; 2 = asleep, easy to awaken; 3 = difficult to awaken; 4 = unresponsive).

**Table 2 jcm-10-00815-t002:** Demographic characteristics for abdominal surgery patients.

	Abdominal Surgery (*n* = 247)		
	Neostigmine (*n* = 47)	Sugammadex (*n* = 204)	Significance *p* Value
ASA (1/2/3/4/E) (%)	5/29/13/0/0 (10/62/28/0/0)	14/138/38/0/14 (7/68/18/0/7)	0.01
Age (mean ± SD)	62 ± 18	60 ± 15	0.42
Height (mean ± SD)	167 ± 10	167 ± 10	1
Weight (mean ± SD)	70 ± 17	73 ± 17	0.2
Hypnotics DES/SEV/PROP (%)	16/31/0 (34/66/0)	67/157/0 (28/72/0)	0.3
ET inhalational agent (%) (mean ± SD)	DES 0.71 ± 0.36 SEV 0.33 ± 0.2	DES 0.74 ± 0.3 SEV 0.27 ± 0.17	0.5 0.03
Duration of surgery (minutes) (mean ± SD)	257 ± 166	318 ± 165	0.02
Temperature (°C) upon arrival in PACU (mean ± SD)	36.6° ± 0.6°	36.6° ± 0.6°	1
Sedation score (0/1/2/3/4) upon arrival in PACU (%)	51/37/7/3/0	48/39/8/3/0	0.3
Length of stay in PACU minutes (mean ± SD)	96 ± 47	91 ± 40	0.45

NA = not analyzed.

**Table 3 jcm-10-00815-t003:** Neuromuscular monitoring characteristics for recovery.

	**Breast Surgery**		**Significance**
	**Neostigmine (*n* = 40)**	**Sugammadex (*n* = 91)**	
T4/T1 after calibration	106 ± 19%	103 ± 28%	0.5
Number of patients with the recorded number of twitches before injection (0/1/2/3/4) %	0/1/1/0/35 0/2.5/2.5/0/87.7	3/30/15/2/31 3/32/16/2/34	*p* < 0.05
Recorded T4/T1 recovery profile after reversal (%) (mean ± SD) Time to TOF ratio 90% mean (min–max)	60 ± 35 77 ± 30 97 ± 22 10 (6–20) (*n* = 22)	46 ± 43 69 ± 43 107 ± 23 2 (1.5–7)	NA NA NA *p* < 0.05
No T4/T1 recorded at extubation	*n* = 4 (10%)	*n* = 18 (20%)	0.2
T4/T1 at extubation	<90% *n* = 14 (35%) >90% *n* =22 (55%)	<90% *n* = 12 (13%) >90% *n* = 62 (68%)	*p* < 0.05
Time to extubation after reversal (mean ± SD)	14 ± 7	13 ± 8	0.4
Time to operating room discharge (mean ± SD)	20 ± 9	19 ± 10	0.5
	**Abdominal Surgery**		
	**Neostigmine (*n* = 47)**	**Sugammadex (*n* = 204)**	
T4/T1 after calibration	119 ± 21%	110 ± 25%	0.02
Number of patients with the recorded number of twitches before injection 0/1/2/3/4 %	0/1/4/4/38 (0//2/8/8/82)	3/92/32/10/45 (1/45/15/5/22)	*p* < 0.05
Recorded T4/T1 profile after reversal (%) time to TOF ratio 90% minute mean (min–max)	58 ± 30 72 ± 23 89 ± 19 9 (5–20)	40 ± 45 70 ± 45 112 ± 20 2 (1.5–8)	NA NA NA *p* < 0.05
T4/T1 recorded at extubation: number (%)	2 (4)	4 (0)	0.05
T4 /T1 at extubation Number (%)	<90% 20 (42) >90% 25 (53)	<90%, 0 >90% 204 (100)	*p* < 0.05 *p* < 0.05
Time to extubation after reversal (minutes) (mean ± SD)	17 ± 10	15 ± 8	0.1
Time to operating room discharge (minutes) (mean ± SD)	23 ± 11	24 ± 11	0.5

NA not analyzed, min = minimum, max = maximum.

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
