# Peer review of "Comparison of the Time to Extubation and Length of Stay in the PACU after Sugammadex and Neostigmine Use in Two Types of Surgery: A Monocentric Retrospective Analysis"

_jcm, 2021, doi:10.3390/jcm10040815_

Round 1

Reviewer 1 Report

Retrospective database study based on GE's Centricity. Well written article, complies with publication standards. As it is a study based on general data, it only allows us to conclude the association between the drugs used and the outcomes found. The article is interesting, but the number of patients that are part of the study is limited to obtain results that can be extrapolated to other studies. Prospective studies would be required to obtain more important conclusions or a larger sample of the retrospective database. Well written study, correct summary and introduction. It details the patients and methods well, the results are well presented, there is no repetition of them in tables and figures. The discussion conforms to what has been published in the literature, correctly relating it to other published articles. It is striking that the use of sugammadex is not adequate, so the work would be more of an internal audit of the use of the drug, which is very useful to assess the efficiency and cost-benefit in real day-to-day situations. For this reason, we believe that its publication is interesting, since it details in a self-critical way some of the situations that should not occur for good pharmacological use motivated by the price and the reuse of vials, the lack of neuromuscular monitoring on many occasions, without observe improvements from the use of sugammadex in the outcomes.

Author Response

Retrospective database study based on GE's Centricity. Well written article, complies with publication standards. As it is a study based on general data,

it only allows us to conclude the association between the drugs used and the outcomes found. The article is interesting, but the number of patients that are part of the study is limited to obtain results that can be extrapolated to other studies

Thank you for your kind comments: we fully agree with the last suggestion and integrated a sentence in the following paragraph

 A recent observational matched cohort study reported that sugammadex administration was associated with a 30–55% reduced risk of major pulmonary complications when compared with neostigmine [15, 16]. We did not search for this complication, mainly because of a lack of power which does not permit to extrapolate our results to other studies.

. Prospective studies would be required to obtain more important conclusions or a larger sample of the retrospective database.

Thank you again with your constructive comments we added the request for further prospective studies in our conclusion

 We believe that generalizing our findings to the general population will be difficult before further prospective studies in these fields are conducted

 Well written study, correct summary and introduction. It details the patients and methods well, the results are well presented, there is no repetition of them in tables and figures. The discussion conforms to what has been published in the literature, correctly relating it to other published articles. It is striking that the use of sugammadex is not adequate, so the work would be more of an internal audit

of the use of the drug, which is very useful to assess the efficiency and cost-benefit in real day-to-day situations.

Thank you for this suggestion we added a sentence in the conclusion:

However, since this was a single center, retrospective, non-randomized analysis, the results cannot be generalized but could be used to at least to improve some of our practice.

 For this reason, we believe that its publication is interesting, since it details in a self-critical way some of the situations that should not occur for good pharmacological use motivated by the price and the reuse of vials, the lack of neuromuscular monitoring on many occasions, without observe improvements from the use of sugammadex in the outcomes.

We also reported the number of files in which we didn’t find any trace of monitoring (8 out 382) which can be explained by multiple factors including no monitoring at all, technical difficulties, the use of other neuromuscular monitoring devices (we do have tof watch and tof scan in the PACU which can be used in the OR in case of failure of the Philips monitor) however there was no any mention of these devices in the anesthesia files.

Reviewer 2 Report

The major consideration is that the significance of your finding in this submission is moderate. Your finding is surely interesting. However, it is difficult to understand which novelty the study add to the current literature.

Even if potentially interesting, the manuscript suffers from many limitations, but many of them are difficult to address:

-Retrospective design must be approached with an appropriate sample size (power analysis is completely lacking)

-What about anaesthesia protocol? Was it standardized? Other variables affecting the Extubation time and Length of Stay in the 2
PACU, including opioids anaesthesia depth monitoring were not included

-The hypothesis was not clarified so one reader can suspect phishing

-The statistical approach is very basic (were the data distribution normal?

- Please provide p values in the tables

Author Response

The major consideration is that the significance of your finding in this submission is moderate. Your finding is surely interesting. However, it is difficult to understand which novelty the study add to the current literature. Even if potentially interesting, the manuscript suffers from many limitations, but many of them are difficult to address:

 This study confirms a better reversal with sugammadex.

-Retrospective design must be approached with an appropriate sample size (power analysis is completely lacking)

Thank you for your suggestion we added a paragraph for sample size calculation as follows:

Sample sizes were chosen with regard to the results of previous investigations [12] assuming a 20% difference in the incidence of residual paralysis between neostigmine and sugammadex by using sample size calculator for comparison of proportion in MedCalc® version 14 (Belgium), the minimum study participants would be 40 for neostigmine and 80 for sugammadex with a power of 95% and α error set of 5%

-What about anesthesia protocol? Was it standardized? Other variables affecting the Extubation time and Length of Stay in the 2
PACU, including opioids anesthesia depth monitoring were not included

Anesthesia protocol is not standardized, however, the absolute majority of patients have propofol for induction, target controlled remifentanil, and inhalational agents for maintenance the bi-spectral index use is also at the discretion of anesthesiologist in charge therefore we couldn’t integrate this parameter for all patients We added a paragraph on these issues thank you for your suggestion

Intensity of ulnar nerve stimulation was calibrated automatically before muscle relaxant injection. Since this was a retrospective assessment, anesthesia protocols were not standardized and were at the discretion of anesthetist in charge. However it always consisted of propofol IV bolus and remifentanil target controlled infusion (RTCI) for induction, and adjusted for maintenance in addition to inhalational anesthesia or propofol IV, taking  also into consideration the presence of not of regional anesthesia (epidural for abdominal surgery and paravertebral block for breast surgery). At skin closure morphine IV 0.1 to 0.15 mg/kg was injected, again in accordance of the presence or not of regional anesthetic technique or infiltration. Postoperative nausea and vomiting protocol is standardized in our institution using dexamethasone and ondansetron intraoperatively followed by droperidol IV if necessary in PACU. There is no specific criteria for the transfer of a patient from operating room to PACU, however all patients need to be extubated unless there is a direct transfer to Intensive care unit with respiratory or hemodynamic support. Patients are discharged from PACU after checking for an acceptable modified Aldrete score including verbal pain and sedation score lower than 2, central temperature > 36°5 and stable hemodynamic and respiratory conditions

-The hypothesis was not clarified so one reader can suspect phishing

We tried to clarify the hypothesis thank you for your suggestion

We hypothesized that the fast recovery of neuromuscular blockade after sugammadex might affect differently extubation time and length of stay of patients in PACU depending on type of surgery with different duration. Our primary objective was to compare the time to extubation, time to exit the operating room, and length of stay in the PACU.

-The statistical approach is very basic (were the data distribution normal?

Yes the software could not perform the t test if the distribution was not normal , we also performed chi square test for all proportionate data

We added also a paragraph for sample size calculations

Sample sizes were chosen with regard to the results of previous investigations [12] assuming an average of 20% difference in the incidence of residual paralysis between neostigmine and sugammadex by using sample size calculator for comparison of proportion in MedCalc® version 14 (Belgium), the minimum study participants would be 40 for neostigmine and 80 for sugammadex with a power of 95% and α error set of 5%.

We

- Please provide p values in the tables

Done

Must be improved

Not applicable

( )

( )

( )

( )

(x)

( )

Reviewer 3 Report

Tables need some adjustments. Some rows are not easily interpretable. For example:

“Number of responses before injection 0/1/2/3/4” presumably reports the number of patients that responded with the given number of twitches after train of four stimulation. This should be cleared.

Many values are reported as “min” and SD, but maybe the authors refer to “mean”.

Units of measurement for all values reported should be specified. 

Table 2, abdominal surgery. ET inhalational agent refers to the end tidal at extubation? Please specify this. 

In table 3, breast surgery “Number of responses before injection 0/1/2/3/4” for patients who received sugammadex are the same both in absolute values and percentage. Please correct the percentage values. 

Line 62. How were missing data completed? Which imputation method was used?

Line 73. Please describe briefly the monitoring device used. Which nerve was stimulated? How does the device detect the twitches (e.g. accelerometer)?

Line 80. What are the criteria for OR and PACU discharge?

Line 124. How is it possible that after a correct dose of sugammadex, the TOFr remains < 90%? Please explain this in the discussion.

Line 137. This is not true. In the study Carron et al showed that sugammadex reduced readiness to discharge and time to discharge both from the OR and from the PACU.

Line 142. “The real time to extubation calculated for each type of reversal might not be totally reliable because, in some patients, the anesthesia providers could have deliberately waited longer in order to avoid the use of sugammadex.” This may be true, but how do the authors explain the equal time to extubation in both groups? I would expect a far shorter time to extubation in the sugammadex group.

Line 151. “Sugammadex might not be necessary sometimes and recovery can be obtained spontaneously”. Note also that some patients were extubated with TOFr less than 90% and try to explain what could have been the cause.

Lines 177-179. This is a conclusion that is not supported by the submitted manuscript, as many patients who were given either sugammadex or neostigmine were extubated with a TOF ratio less than 90%. This means that they had residual paralysis.

Author Response

Tables need some adjustments. Some rows are not easily interpretable. For example:

“Number of responses before injection 0/1/2/3/4” presumably reports the number of patients that responded with the given number of twitches after train of four stimulation. This should be cleared.

Thank you for your suggestion, we changed the description of this row as suggested

Many values are reported as “min” and SD, but maybe the authors refer to “mean”.

We apologize for the confusion, we made corrections

Units of measurement for all values reported should be specified.

 Done

Table 2, abdominal surgery. ET inhalational agent refers to the end tidal at extubation? Please specify this. 

We explained it at the end of the table

In table 3, breast surgery “Number of responses before injection 0/1/2/3/4” for patients who received sugammadex are the same both in absolute values and percentage. Please correct the percentage values. 

Done

Line 62. How were missing data completed? Which imputation method was used?

We added a paragraph for missing data thank you for your suggestion.

The outliers were checked on the anesthesia report and corrected oromitted. The missing data (weight, height for example) were extracted from the patient’s medical report. 

Line 73. Please describe briefly the monitoring device used. Which nerve was stimulated? How does the device detect the twitches (e.g. accelerometer)?

Line 80. What are the criteria for OR and PACU discharge?

. There is no specific criteria for the transfer of a patient from operating room to PACU, however all patients need to be extubated unless there is a direct transfer to Intensive care unit with respiratory or hemodynamic support. Patients are discharge from PACU after checking for an acceptable  modified Aldrete score including verbal pain  and sedation score lower than 2, central temperature > 36°5 and stable hemodynamic and respiratory conditions

Line 124. How is it possible that after a correct dose of sugammadex, the TOFr remains < 90%? Please explain this in the discussion.

Before injecting Sugammadex, it is necessary to ensure that there is at least one response to ulnar nerve stimulation. Otherwise, a post-tetanic count should be performed to adjust the dosage of sugammadex. This control has not always been performed and it is possible that the injected dose of sugammadex was not sufficient to ensure complete reversal

Line 137. This is not true. In the study Carron et al showed that sugammadex reduced readiness to discharge and time to discharge both from the OR and from the PACU.

Thank you for your suggestion we deleted “partially in accordance.

Carron et al. [7], determined that sugammadex decreased readiness time and time to discharge both from operating room and PACU for morbidly obese patients

Line 142. “The real time to extubation calculated for each type of reversal might not be totally reliable because, in some patients, the anesthesia providers could have deliberately waited longer in order to avoid the use of sugammadex.” This may be true, but how do the authors explain the equal time to extubation in both groups? I would expect a far shorter time to extubation in the sugammadex group.

When sugammadex was injected, the expired concentration of halogenated gas was relatively high especially after breast surgery. It is possible that the extubation delay was influenced more by the return to consciousness t

 Moreover it is possible that time to extubation was more influenced by the awakening rather than muscle relaxant’s recovery as expired concentration of inhalational anesthetics was relatively high han the muscle relaxant reversal.

? I would expect a far shorter time to extubation in the sugammadex group.

We also expected this event however it appears in our cohort of patients that time to extubation is not only dependent to TOF ratio and other factors plays their part basically the return to consciousness.

Line 151. “Sugammadex might not be necessary sometimes and recovery can be obtained spontaneously”. Note also that some patients were extubated with TOFr less than 90% and try to explain what could have been the cause.

The necessity of   monitoring TOF ratio to manage muscle relaxation before extubation is  still a concept that is not acquired by all anesthesia providers and it is possible that some neglect to take into account the values of monitoring before extubation. The values of T4/T1 ratio are measured and displayed automatically on the monitor screen may be  the team does not see the numbers at the right time  or other factors such as extreme agitation of the patient might also affect judgement. 

In addition, a very small number of patients (2%) did not have any trace of monitoring without any plausible explanation however putative technical problem cannot be ruled out

Lines 177-179. This is a conclusion that is not supported by the submitted manuscript, as many patients who were given either sugammadex or neostigmine were extubated with a TOF ratio less than 90%. This means that they had residual paralysis.

We agree with your suggestion , however we meant  that the combination of monitoring and sugammadex was at lower risk of residual paralysis in comparison to NMB monitoring and neostigmine. We tried to improve this statement as follows;&

To avoid residual paralysis, it is necessary to monitor all patients until extubation time regardless of the administration of reversals. This is true for, before administration  and before extubation with different criteria depending on  which reversal  is used.

.

Our results somewhat confirm previous findings that the use of intra-operative quantitative NMB monitoring and sugammadex is probably associated with a lower incidence residual paralysis [18] in comparison to NMB monitoring and neostigmine

Round 2

Reviewer 3 Report

I believe that your work was greatly improved by the last revision. The last thing that i suggest, despite not being mandatory for publication, is to justify the fact that there are many differences between groups in terms for example of surgery duration.

This is absolutely understandable, as the study design is retrospective and possibly surgery duration was one of the criteria that was taken into consideration by the attending anesthesiologist to choose between sugammadex and neostigmine.

The authors should probably discuss this in the paper, as this might also increase the scientific soundness of the work.

Author Response

Dear editor 

Thank you for your suggestion , we added a sentence in the discussion section as asked 

.” Moreover, we also speculate that the duration of surgeries were also taken into consideration by the attending anesthesiologist to choose between sugammadex and neostigmine.  
